# Quantifying uncertainty in wave attenuation by mangroves to inform coastal green belt policies
Bregje K. van Wesenbeeck [1,2] ✉, Vincent T. M. van Zelst [1,2], Jose A. A Antolinez [1] & Wiebe P. de Boer [2]

The capacity of mangroves to reduce coastal flood risk resulted in legislation for mandatory widths of mangrove greenbelts in several countries with mangrove presence. Prescribed forest widths vary between 50 and 200 m. Here, we performed 216,000 numerical model runs informed by realistic conditions to quantify confidence in wave reduction capacity of mangroves for wind and swell waves. This analysis highlights that tidal flat areas fronting mangrove forests already account for 70% of reduction in wave heights. Within mangrove forests that are below 500 m wide, wave dissipation is strongly dependent on local water levels, wave characteristics and forest density. For forest widths of over 500 m, which constitute 46% of global coastal mangroves, around 75% or more of the incoming wave energy is dissipated. Hence, for relying on mangroves to dampen shorter waves, a new standard should be adopted that strives for mangrove widths of 500 m or more.

Globally, an increasing number of people and assets is vulnerable to coastal flooding[1,2]. This number is expected to rise with population growth and sea level rise, especially in tropical regions[3]. Mangroves occupy around 10% of global coastlines[4] and occur specifically in tropical and subtropical zones. Global flood risk reduction provided by mangroves has received considerable attention, and integration of mangroves in flood risk reduction and climate change adaptation strategies can result in costs savings for coastal infrastructure[4–6]. Mangroves are estimated to reduce coastal flood risk from surges and waves by 250 million USD annually[5] and to decrease future flood risk by 8.5%[6]. Mangroves contribute to coastline stabilization and reduction of coastal flooding by damping incoming wind and swell waves[7,8], by reducing surges[5], and by enhancing sedimentation and reduce erosion[9]. They are expected to contribute most to risk reduction along rural coastlines where there is more space and mangrove forests are more extensive. In areas where there is less space and where populated lands are low-lying, mangrove forests in front of sea walls can considerably reduce the costs for sea wall construction[4]. The implementation potential for such hybrid defenses, is considered high as this may also apply for urbanized coastlines[10,11].

Despite the valuable coastal protection that mangroves provide, these forests are still replaced for fish and shrimp farming and urban development and are unsustainably exploited for firewood and timber[12]. Across the globe countries have policies that protect mangroves and that strive to limit mangrove cutting for aquaculture or urban expansion[7]. In these policies often greenbelt zones along coasts and rivers are advised or are mandatory to protect coasts from waves, erosion, and salinity intrusion[13,14]. These greenbelt zones vary in recommended widths from 50 to 1500 m[7]. Based on previous studies a width of approximately 100 meters is minimally required from the perspective of limiting hydraulic forces[7,15,16]. However, the exact width for effective hazards mitigation is influenced by the type and strength of the hazard, such as water levels and waves, and by characteristics of the mangrove greenbelt, such as tree density and tree diameter[17–19]. Multiple studies highlight the sensitivity of wave attenuation by mangroves for a range of abiotic and biotic characteristics, such as tree density, greenbelt width, and wave heights, that can differ between sites and over time[8,20]. However, except for a model and experimental exploration on the effects of different age classes of mangroves[19,21], a systematic exploration of the sensitivity of wave attenuation through mangrove forests to these characteristics is yet lacking. This makes it difficult to generalize results for use in coastal protection policies and designs. In addition, most studies[23] use numerical models that are calibrated with relatively small waves (i.e., significant wave heights up to ~0.7 m and peak wave periods up to ~10 s) and low water levels (i.e., up to ~2 m)[22–24]. However, vegetation may be less effective to attenuate waves under storm conditions with high water levels and higher waves[25–27]. A systematic quantitative analysis of wave attenuation for a wide range of biotic and abiotic characteristics, including more extreme conditions, and what this implies for effective mangrove greenbelt widths is currently lacking.

[1]Delft University of Technology, Faculty of Civil Engineering and Geosciences, Delft, The Netherlands. [2]Deltares, Delft, The Netherlands. ✉e-mail: B.K.vanWesenbeeck@tudelft.nl

Here, we executed a probabilistic assessment of wave attenuation by mangrove greenbelts and fronting tidal flats based on global data in combination with numerical models. First, data mining on global data of vegetation widths, bed levels, wave conditions, and water levels was undertaken for the parameterization of global mangrove environments. We then used a Maximum Dissimilarity Algorithm (MDA) to select 1000 representative parameter combinations. We combined these with a range of values for key mangrove tree characteristics extracted from an extensive literature survey. This resulted in a total of 216,000 numerical simulations. To narrow model uncertainty the wave dissipation by vegetation parameterization in the numerical model was calibrated with experiments of a real scale forest under extreme conditions[26]. The output from simulations was used to characterize wave attenuation classes using the proportion of wave reduction taking place over the foreshore and within the mangrove forest itself. Resulting classes are contrasted on a global map, indicating in what parts of the world mangroves contribute with high and low certainty to reducing wave action.

## Methods

To assess uncertainties in wave attenuation by mangrove greenbelts induced by biotic and abiotic factors, we executed a large set of numerical experiments on characteristic global mangrove environments. Parameterization of mangrove environments is carried out using a set of morphological, hydrodynamic, mangrove forest and tree characteristics (Fig. 1). For selection of modeling conditions, we applied data mining on global data sources and executed a literature study on mangrove characteristics[4]. Synthetic 1D coast-normal transects were set-up for modeling runs. Key elements of the numerical set-up include the parameterization of the vertical

distribution of mangrove biomass and the calibration of wave dissipation by woody vegetation using experiments of a real scale forest under extreme conditions[26].

### Transects

We used globally available transects of 8 km length (~4 km seaward and 4 km landward) oriented shore-normal to the OpenStreetMap coastline[28] with an alongshore spacing of ~1.1 km[4]. For the current study only a subset of transects in the tropics is analyzed. This subset contained only transects vegetated with mangroves at non-sheltered locations with realistic hydrodynamic conditions. Transects at sheltered locations were removed, as the offshore wave data (from ERA-Interim) are considered unreliable for these locations. A transect was marked as sheltered if it intersected the coastline more than once (in cross-shore direction). This selection resulted in the inclusion of 15,773 transects that are vegetated with mangroves. These transects cover a wide range of vegetation widths, foreshore depths, water levels and wave heights.

### Bathymetry and topography

The domain of the transect was split into three parts: (1) offshore, (2) foreshore and (3) mangrove greenbelt (Fig. 1) with the aim to derive representative coast-normal transects for non-sheltered coastal geomorphological settings. The bathymetric profile of the transects started at deep water ($-100$ m $+$ MSL) with a slope of 1:20 (part 1). Next, the depth at the start of the foreshore ($Fs_{z0}$) was derived from global data (Table 1)[4]. We implemented three foreshore slopes based on literature: 1:500, 1:750 and 1:1000 [16,29]. For the area between the start of the foreshore and the start of the forest (part 2), a minimal threshold width of 50 m was applied. For the

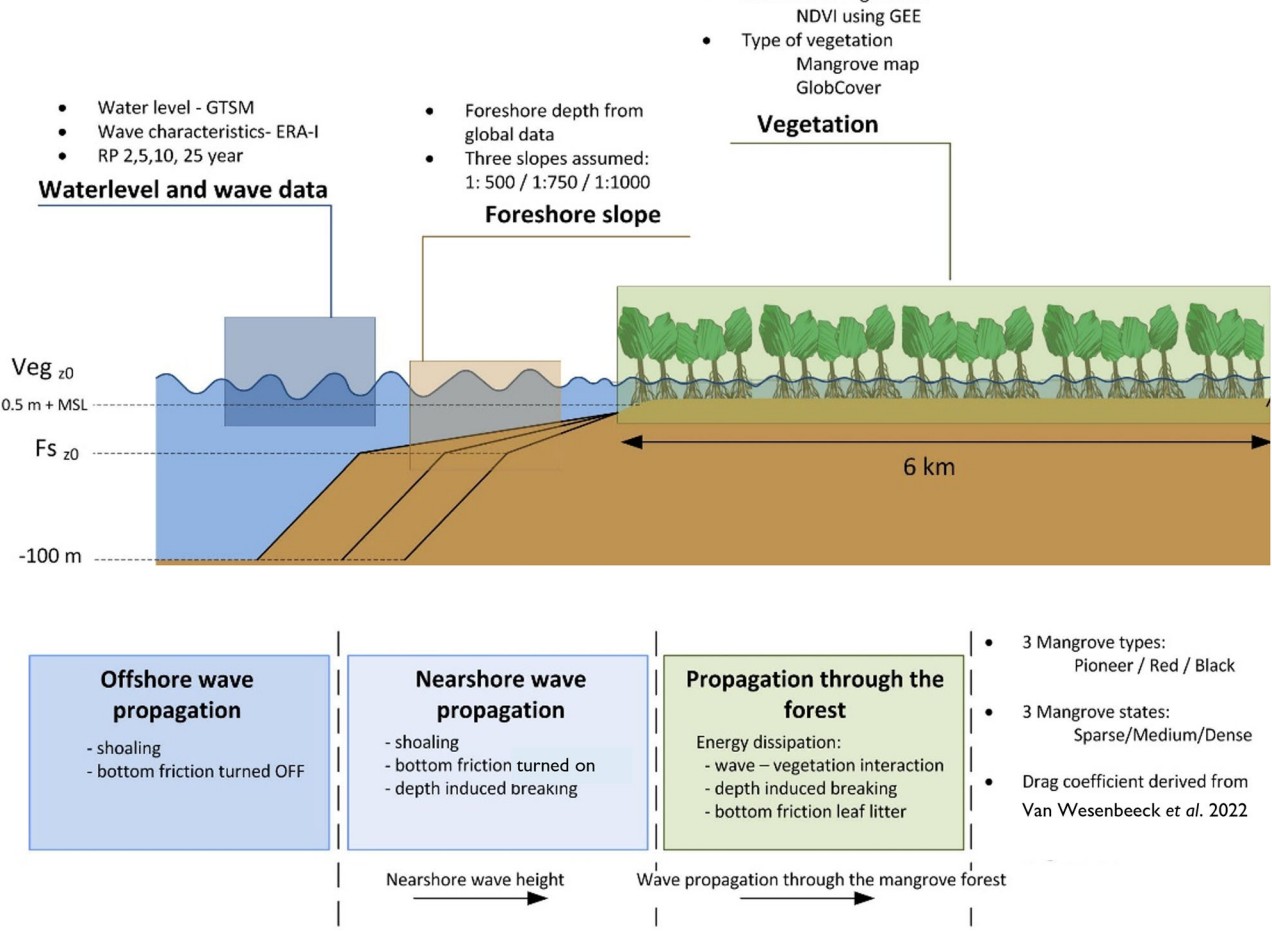

**Fig. 1 | Representation of model domain, model conditions, and computational steps.**

**Table 1 | The 5%, 50% and 95% percentile for all input parameters from global data for 15773 coastal locations**

| Variable name | 5% percentile | 50% percentile | 95% percentile |
|---|---|---|---|
| Extreme water level (RPs 2, 5, 10, 25) | 0.7 m | 1.6 m | 3.5 m |
| Significant wave height (RPs 2, 5, 10, 25) | 1.7 m | 2.8 m | 5.0 m |
| Bed level at start of the foreshore ($Fs_{z0}$) | −2.6 m +MSL | −1.1 m +MSL | −0.6 m +MSL |
| Mangrove belt width | 50 m | 400 m | 3335 m |

mangrove greenbelt (part 3) an elevation ($Veg_{z0}$) of 0 m +MSL and 0.5 m +MSL was assumed which is based on average distribution of mangroves with respect to inundation frequency[30].

### Hydrodynamic data
Water level data were available for nine return periods (RPs: 2, 5, 10, 25, 50, 100, 250, 500, 1000 years) from the Global Tide and Surge Reanalysis[31]. However, for socioeconomic relevance only water levels with return periods (RPs) of 2, 5, 10, and 25 years were included to avoid too extreme and unrealistic values to influence the overall analysis (Table 1). Offshore wave conditions were based on a reanalysis of ERA-Interim[32] using data from 1979 until 2017. This dataset included both wind and swell waves, also under storm conditions. An automated Peak Over Threshold extreme values analysis was applied to determine the significant wave height and the peak wave period values for return periods of 2, 5, 10, and 25 years (Table 1). We excluded transects with a wave steepness exceeding the wave steepness criterium $k_pH_s/2 > 0.142$[33], where $k_p$ is the wave number derived from the peak wave period $T_p$ and $H_s$ the significant wave height).

### Mangrove cover
The mangrove greenbelt width was based on vegetation presence derived from earth observation data between 2013 and 2017 combined with a global mangrove inventory dataset[34], complemented with GlobCover v2.2[35]. Vegetation width was calculated by the sum of vegetated transects cells on 8 km long cross-shore transects (Table 1). The applied transects cell size of 25 m resulted in exclusion of mangrove belts below this threshold.

### Mangrove schematization for numerical modeling
Frontal surface area for red mangroves (*Rhizophora Sp.*), black mangroves (*Avicennia Sp.*) and pioneer mangroves was determined on the following: 1. literature for stems and roots of red and black mangroves, 2. scaling of biomass for canopies due to lack of field data (Narayan 2009) and 3. assessment of photographs in case of pioneer vegetation. Based on these data, frontal surface area for each species was determined (Fig. 2) and included in the numerical wave model SWAN[36,37]. Numerical simulations were executed with three different surface areas for each species to gain insight into the effect of varying density in trees, stems, and branches within a forest. Medium surface area relative to water level is shown in Fig. 2. Note that red and black mangroves are generally emergent, and that pioneer vegetation is in certain cases fully submerged. Sparse and dense forests were represented by a deviation of ±20% of the total frontal surface area (see Supplementary Materials S1).

### Selection of representative clusters
The subset of 15,773 locations with mangrove cover contained the following information: location (longitude, latitude), vegetation width, depth at the start of the foreshore, and hydrodynamic conditions, such as significant wave height ($H_s$), peak period ($T_p$) and water level) for four return periods (RPs 2, 5, 10, 25). We used the Maximum Dissimilarity Algorithm (MDA) to select 1000 representative parameter combinations that maintain the dependency and correlation between the natural forcing, the depth at the start of the foreshore, and the mangrove greenbelt width. The concept of MDA is to sequentially select a reduced number of model scenarios that cover the entire multivariate data space. In contrast with other data mining techniques, the MDA subset keeps the representativeness of the original

dataset while exploring less dense areas of the multivariate space[38]. A low number of selected representative locations will result in a poor characterization and a high number of locations will result in running simulations for very similar conditions which is computationally cost-ineffective. Next, we combined all 1000 MDA representative conditions, with nine mangrove structure features (i.e., type and density), three different foreshore slopes, and two bed level values at the start of the forest. Because we simulated each of the four return periods (RPs 2, 5, 10, 25), it resulted in 216 (9 × 3 × 2 × 4) combinations per MDA condition (Table 2). In total 216,000 model simulations were executed with a resolution of 5 m along the transect. For each model simulation there were 1200 output points within the 6 km forest. Hence, in total 259,000,000 data points were derived for this analysis. For post-processing, we focused on the first 2 km forest, including 86,400,000 data points in total.

### Numerical model set-up
For synthetic model simulations the numerical model 1D-SWAN was used[36]. The model domain contained of 8 kilometers long transects and was split into three parts: (1) offshore, (2) foreshore, and (3) mangrove greenbelt (Fig. 1). Each calculated greenbelt provided results for a whole range of greenbelt widths by outputting data at different distances into the forest. A uniform grid cell size of 5 meters was used.

### Wave dissipation model
The spectral wave model SWAN (*Simulating Waves Nearshore*)[36] was used. This model includes the wave attenuation by vegetation formulation of Mendez and Losada (2004)[39], which was implemented in SWAN accounting for vertical layering by Suzuki et al. (2012)[37]. SWAN was run in its 1D stationary mode, in a Cartesian and regular computational grid. The mangrove forest was modeled by accounting for 7 vertical layers of vegetation (expressed as frontal surface area), which were assumed to be uniform along the forest length. SWAN is based on the bulk wave dissipation (integrated over all wave frequencies), which depends on the incoming wave energy, relative water depth and vegetation characteristics (Eq. 1)[40]:

$$\langle \varepsilon_v \rangle = \sum_{i=1:7} \frac{1}{4\sqrt{2\pi}} \rho \widetilde{C_D} \left( \frac{gk}{2\sigma} \right)^3 f_i \frac{\left( \sinh^3 k\alpha_i h - \sinh^3 k\alpha_{i-1}h \right) + 3\left( \sinh k\alpha_i h - \sinh k\alpha_{i-1}h \right)}{3k \cosh^3 kh} H_s^3$$

(1)

Where: $\langle \varepsilon_v \rangle$ is the averaged wave energy dissipation due to vegetation, $\widetilde{C_D}$ the bulk drag coefficient, $g$ the gravitational acceleration constant, $k$ the mean wave number, $\alpha$ the water depth covered by vegetation for layer i, $h$ the water depth, $H_s$ the significant wave height, and $f_i$ the total frontal width of vegetation per surface area for layer i, which is equivalent to the generally used $b_{v,i}N_{v,i}$.

The numerical parameterization of wave attenuation by vegetation requires estimating drag by vegetation characteristics under different hydraulic conditions. This is often obtained from a relationship between bulk drag coefficient ($C_D$) for woody vegetation and a dimensionless parameter[40], for example using the Keulegan-Carpenter (KC) number[41]. The KC number is a dimensionless parameter describing the relative importance of the drag forces over the inertia forces. However, a KC-$C_D$ relationship for wave-mangrove interaction under extreme hydrodynamic conditions is not available. Therefore, we applied the calibrated KC-$C_D$

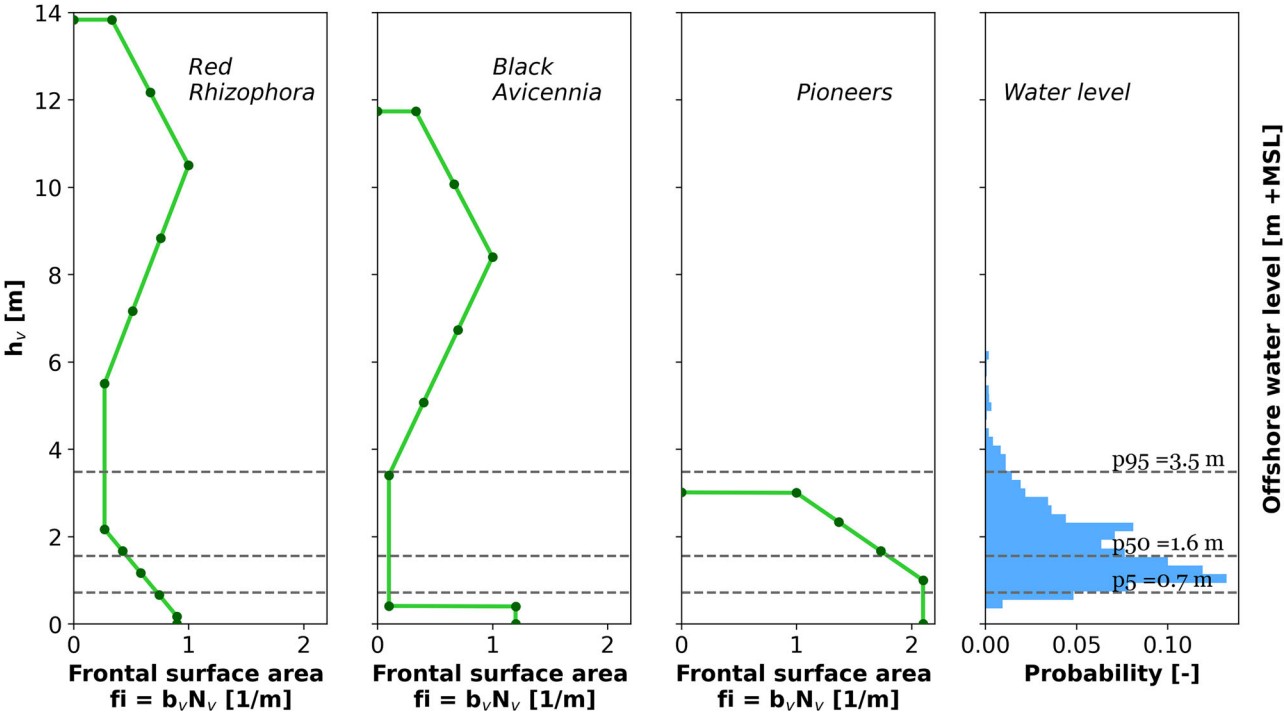

**Fig. 2 | Mangrove schematization of frontal surface area (fi), based on stem diameter (bv) and stem density (Nv) for each height (hv) based on literature (see S1 for more info), for red and black mangrove, and on pictures, for pioneer mangroves, and water level distribution for RPs = 2, 5, 10, 25 years from global data, n = 15773.**

## Table 2 | Input model scenarios

| Variable | # Scenarios | Scenario characteristics | | | |
|---|---|---|---|---|---|
| Mangrove types | 3 | Pioneer | Red | Black | |
| Mangrove density | 3 | Sparse | Medium | Dense | |
| Foreshore slope | 3 | 1:500 | 1:750 | 1:1000 | |
| Bed level start forest | 2 | 0.0 m +MSL | 0.5 m + MSL | | |
| Return periods | 4 | 2 yrs | 5 yrs | 10 yrs | 25 yrs |

relationship derived for willows[26] to simulate the wave attenuation in mangrove greenbelts. The KC number for the mangroves was computed using the Mazda length scale[42], where the amplitude of the orbital velocity is based on the depth limited wave height just in front of the mangrove forest.

Different $C_D$ values were calculated for the three vertical sections of the mangroves: the roots, the trunk, and the canopy. We applied the Collins formulation for a spatially varying bottom friction[43]. Offshore the bottom friction was turned off to prevent unrealistic wave dissipation before the wave entered the synthetic tidal flat profile. We did not account for additional wind growth effects on waves. On the tidal flat foreshore the default Collins factor of 0.015 was applied. Inside the forest this factor is multiplied by 1.4 to account for leaf litter and root structure. We applied a JONSWAP wave spectrum with a peak enhancement factor of 3.3 and directional spreading of 30 degrees. The following other numerical settings were applied: relative change of significant wave height ($H_s$) and mean average wave period ($T_{m01}$) of 0.005, relative change of $H_s$ with respect to mean of 0.01, relative change of $T_{m01}$ with respect to mean of 0.005, percentage of wet points of 99.5 % and a maximum number of iterations of 50. Histograms of values for each parameter in relation to the amount of wave reduction can be found in the supplementary materials (Figure S1).

### Reporting summary

Further information on research design is available in the Nature Portfolio Reporting Summary linked to this article.

## Results

Figure 3 shows sensitivity of wave heights to dissipation on the foreshore and mangrove forest for all aggregated simulations ($n = 216,000$). Noticeable is the rapid decline in significant wave heights over the shallower foreshore due to bottom friction and depth-induced wave breaking (Fig. 3). During the highest wave heights (defined by the 95 percentiles of the waves simulated, P95, red line in Fig. 3) wave dissipation over the foreshore due to bottom friction and wave breaking can account for up to 70% of the total wave height reduction. For lower incoming waves (the 5 percentile, P5, green line in Fig. 3) total reduction amounts up to 80%.

Similarly, looking at the wave attenuation within the forest for all aggregated simulations (Fig. 4), reveals that most wave energy is attenuated in the first 500 m of the forest. Here, the median value (50 percentile, P50) of the wave attenuation rises to 90% at 500 m (Fig. 4). The confidence band (distance between the P95 and P5 line) decreases rapidly from around 50%-point for a forest width of 25 m to below 10%-point for a greenbelt width of 2000 m. The large uncertainty band for smaller mangrove belt widths indicates that the exact amount of wave attenuation for these widths varies largely depending on other parameters, such as incoming wave height and tree density. The slope of the lines indicates the rate of wave energy dissipation through the forest. The median value of wave attenuation within the first 100 m of forest is 62 % (Fig. 4). In the next 400 m the reduction increases with 28%-point to 90% (i.e., after 500 m), and in the subsequent 500 m wave height further decreases with 4.8%-point to 95% (i.e., after 1000 m). Hence, the relative amount of energy reduction per meter gets lower once the wave already traveled through the forest for a considerable distance and is already partly dampened, which is confirmed by prior field measurements and modeling[44–47].

For all mangrove locations ($n = 15773$) we visualized classes of mangrove width that we distinguished in Fig. 4 on a global map with indicators for each IPCC6 region (Fig. 5). Most locations fall within the Southeast Asia region ($n = 4932$). Notably, for most regions the width of present greenbelts falls within the green and yellow class implying that mangrove widths exceed 100 m, which means that they also play a significant role in wave

**Fig. 3 | The bandwidth of wave propagation relative to distance from the start of the forest for 216,000 simulations from offshore to within the mangrove forest.** X-axis shows the relative distance from the start of the forest (%) with −100% being the offshore model boundary, 0% the transition of the bare foreshore (tidal flat) into the mangrove forest and +100% the end of the mangrove forest. Y-axis depicts significant wave heights (Hs). Purple, yellow and green lines represent the 5, 50, and 95 percentile wave heights respectively.

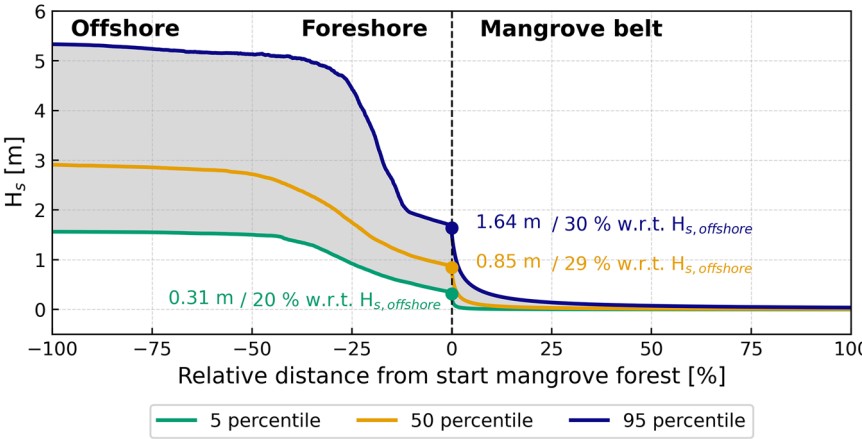

**Fig. 4 | Percentage of wave reduction for a certain width of mangrove forest based on 86,400,000 data points with different mangrove types, densities, foreshore slopes and return periods.** The x-axis shows the percentage of wave attenuation and the y-axis the width of the mangrove green belt. Dotted, solid, and intermitted lines represent the 5, 50, and 95 percentiles of the wave reduction. Purple, yellow, and green areas represent uncertainty classes based on total band width of wave attenuation (approximately >50%-point, 20–50%-point and 10–20%-point).

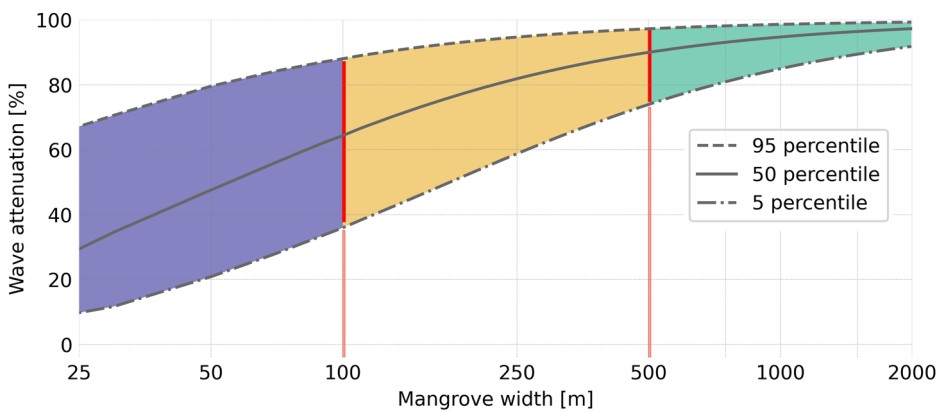

attenuation. The amount of red transects, where mangroves are between 0 and 100 m are more often found in subtropical regions where mangroves only occur limitedly due to temperature constraints.

## Discussion

To fully explore the potential of mangroves to mitigate coastal flooding and erosion, there is a strong need for systematic model analysis and sensitivity testing to explore levels of certainty that can be translated to design guidelines. In this study, we have systematically explored the uncertainty in wave attenuation by combinations of tidal flats and mangroves for a wide range of biotic and abiotic factors. This is relevant for around 10% of global coastlines, which are occupied by mangroves. Our study highlights that both the tidal flat dimensions and the vegetation structure are relevant for the wave attenuation. Tidal flats occur all over the globe with hotspots of occurrence in Asia and the Americas[48]. Our results show that tidal flats alone account for more than half of the attenuation of incoming waves due to bottom-friction and wave breaking in front of the vegetation. Mangroves are generally found in sheltered coasts and bays and do not occur along open and wave dominated coasts. Hence, for mangroves to be able to colonize and establish waves should be attenuated by fronting coral reefs, sand ridges or gradually sloping tidal flats. Several recent studies illustrate that these systems are mutually dependent[49,50] and the importance of foreshores for wave attenuation in front of mangrove forests has been demonstrated in several case studies[51–53] The current study provides global evidence for the importance of tidal flats to ameliorate wave conditions for mangrove forests. However, mudflats are declining globally, which may result in cascading erosive effects on mangroves and marshes, especially under sea level rise[48,53].

Similarly, many local studies demonstrated wave attenuating capacities of mangroves[22,24]. In addition, numerical modeling studies provide insight in the effectiveness of different mangrove types in attenuating waves[54].

However, a more comprehensive assessment that allows extrapolation to other areas of the generic capacity of mangrove to attenuate waves was yet lacking. Here, we showed that within mangrove greenbelts, wave attenuation increases with the width of the forest. Our analysis revealed that mangrove green belt widths over 500 m can effectively attenuate wind and swell waves (by over 75% compared to the waves at the start of the forest) independent of other conditions. Below 500 m, parameters, such as incoming wave heights, water levels and mangrove densities largely determine the amount of wave reduction, hence, the uncertainty is larger. This is relevant for 54% of the mangrove greenbelts around the globe, which are less than 500 m wide.

Existing guidelines and policies for mangrove greenbelts often target a width of 100–200 m of mangrove forest or less. Our results suggest that this is relatively small, even without considering the most extreme conditions (e.g., tropical storms and cyclones), low-frequency waves, and possible failure mechanisms of mangrove greenbelts. Nevertheless, wider green belts in combination with gradually sloping tidal flats are more effective with more confidence to dampen waves. However, in many places these cannot realistically be conserved or restored due to limits in coastal topography on the front and encroachment of anthropogenic activities in the back. Hence, for smaller forest widths it needs to be realized that wave attenuation strongly depends on a whole range of other factors. Some generic trends are confirmed by our results, such as wave attenuation tends to be smaller for larger water depths, larger wave conditions, and less dense greenbelts. In addition, our results confirmed that young (pioneer) mangrove greenbelts of relatively small width can be effective in attenuating waves under non-extreme conditions in shallow water depths, where their roots and branches obstruct the waves[19]. This suggests that restoration of mangroves can relatively quickly benefit coastal protection. However, pioneer mangroves are less effective in attenuating waves under higher water levels when dense root

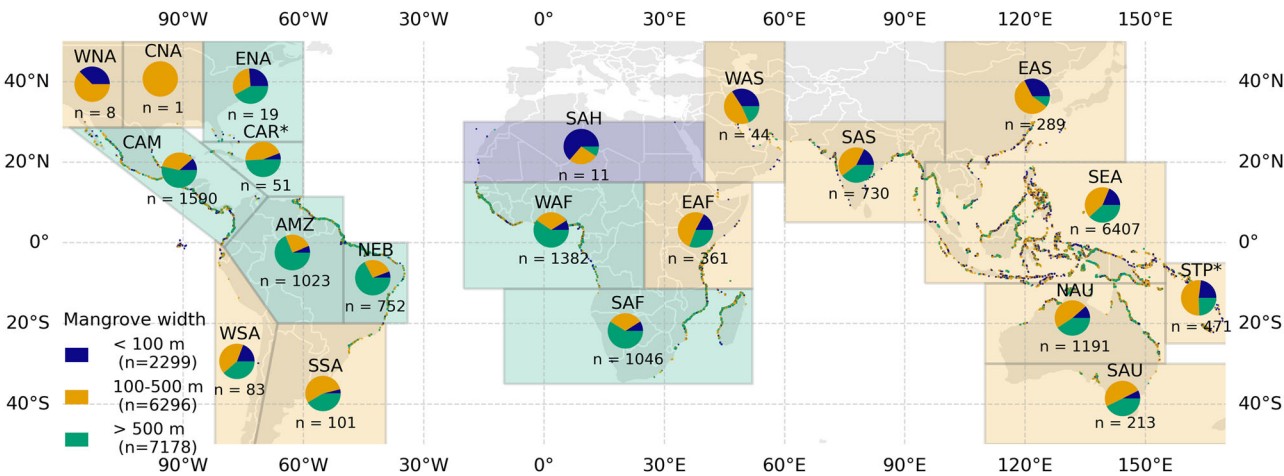

**Fig. 5 | Global map of mangrove width classes in purple 0–100 meters, in yellow 100–500 meters, in green everything larger than 500 meters.** Pie charts show distribution of different classes for each IPCC6 region. The legend shows the total number of transects within each class. The total number of transects is $n = 15.773$.

systems and the canopy are submerged. Furthermore, it is unknown to what extent pioneer mangroves can withstand extreme conditions. Hence, older trees or a mixed forest are more likely to be effective as coastal protection buffer under extreme conditions.

The current study focusses on the role of mangroves on attenuation of waves. However, storm surges, infra-gravity waves and typhoon conditions are not included. Mangroves are less effective in lowering long-period storm surges[55]. Hence, mangrove greenbelts by themselves are not likely to prevent flooding of the hinterland completely. a combination of mangroves fronting a water retaining structure (e.g., a levee), so-called hybrid solutions, could be considered. To fully explore the potential of fully natural or hybrid solutions that combine vegetation with sea walls, there is a strong need for systematic model analysis and sensitivity testing as performed in the current study to explore levels of certainty that can be translated to design guidelines. For local application of hybrid intervention to mitigate flood risk, including mangrove forests, exploration of the importance and sensitivity of specific local conditions on the exact amount of wave attenuation is needed. In addition, there are no design guidelines to capture mangrove characteristics in numerical models for local combinations of mangroves and dikes, embankments or sea walls.

An exploration of the presence of mangroves along global coastlines illustrates that in many areas significant mangrove widths can still be found. These mangrove areas play a crucial role in reducing wave impacts and reducing erosion of the coastline. Hence, conservation of larger stretches of forest is strongly recommended to avoid coastline destabilization. Our results also put global mangrove restoration efforts in perspective. It is illustrated here that mudflats are essential to reduce wave heights in front of mangrove forest and are not the best place for mangrove restoration. To restore mangrove forests, restoration methods that find space in the back where mangrove areas have previously been converted to other land uses, such as fish and shrimp ponds, are better suitable. Finally, next to wave attenuation properties of mangroves, ecological resilience should be equally considered, implying that in guidelines and polies, desired mangrove width should also be informed by ecosystem health and resilience of mangrove habitats.

## Data availability
Input parameters and output data of model runs can be found here: The Importance of Mangrove Greenbelt width to Reduce Storm Waves[56] File B includes hydrodynamic inputs for the SWAN modeling and can be opened using the Python package PANDAS. File A is a comma separated file containing the data behind the percentile lines of the relationship between wave attenuation by mangroves as function of the mangrove width used to constitute Fig. 4. File C is a NetCDF file containing the Significant wave

height (Hs) outputs within the mangrove green belt corresponding to the scenarios specified in Table 2 and is used constitute Figs. 4, 5 and S2. File D (csv) wave propagation relative to distance from start of the mangrove forest (Fig. 3). Files E (csv) mangrove schematizations as presented in Fig. 2.

## Code availability
All general-purpose software packages that we used are open source: Python 3.10.13 (https://www.python.org), NumPy (http://www.numpy.org/), GeoPandas (http://geopandas.org/), Xarray (http://xarray.pydata.org/en/stable/), SciPy (https://www.scipy.org/), Shapely (https://pypi.org/project/Shapely/). Maps were created using Cartopy (v0.22.0. Met Office UK. https://pypi.python.org/pypi/Cartopy/0.22.0) and Matplotlib v3.8.2. The numerical models used in this study are available at: (SWAN) http://swanmodel.sourceforge.net/. The software written specifically for this project is available from the corresponding author on reasonable request.

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

## Acknowledgements

This work is executed under various projects. It builds on the work done under the FAST project (Foreshore Assessment using Space Technology), which was funded by the European Union's (EU) Seventh Framework Programme (FP7) for research, technological development, and demonstration under grant agreement number 607131. Large parts were executed as part of the JIP Woody consortium. We therefore thank van Oord, Boskalis, Rijkswaterstaat, World Wildlife Fund, Stowa, Wetlands International, Ecoshape, and VP Delta for their contributions to this research. We thank Merijn Janssen for the literature study on mangrove parameters that were used in this work. The work was finalized under the research program WOODY, number 17194, which is financed by the Dutch Research Council (NWO) via the Open Technology Program. We thank two anonymous reviewers and Ken W. Krauss for providing constructive comments and input that improved our manuscript.

## Author contributions

B.K.vW. developed the idea and conceptual method and wrote the manuscript. V.T.M.vZ. performed model runs, ran, and wrote the (post) processing pipeline and produced all figures. J.A.A.A. executed data mining and contributed to writing the manuscript. W.P.dB. participated in conceiving the conceptual method, supervised the modeling work, and contributed to writing the manuscript.

## Competing interests

The authors declare no competing interests.
