## [Transparent Peer Review file · Communications Earth & Environment]

Reduction of Uncertainty in Wave Attenuation by Mangroves to Inform Coastal Green Belt Policies

Corresponding Author: Dr Bregje van Wesenbeeck

Version 0:

Decision Letter:

Dear Dr van Wesenbeeck,

Your manuscript titled "Quantifying confidence in the effective width of mangrove green belts to reduce storm waves" has now been seen by 3 reviewers, and we include their comments at the end of this message. They find your work of interest, but some important points are raised. We are interested in the possibility of publishing your study in Communications Earth & Environment, but would like to consider your responses to these concerns and assess a revised manuscript before we make a final decision on publication.

We therefore invite you to revise and resubmit your manuscript, along with a point-by-point response that takes into account the points raised. Please highlight all changes in the manuscript text file.

Please submit your point-by-point responses as a separate file, distinct from your cover letter where you can add responses to the Editors' comments that you do not want to be made available to the reviewers. Word files are preferred. We recommend that any figures, tables or graphs that are included in the response to reviewers are also included in the main article or Supplementary Information.

Please use the following link to submit your revised manuscript, point-by-point response to the referees' comments (which should be in a separate document to any cover letter), a tracked-changes version of the manuscript (as a PDF file) and the completed checklist:

Link Redacted

We hope to receive your revised paper within six weeks; please let us know if you aren't able to submit it within this time so that we can discuss how best to proceed. If we don't hear from you, and the revision process takes significantly longer, we may close your file. In this event, we will still be happy to reconsider your paper at a later date, as long as nothing similar has been accepted for publication at Communications Earth & Environment or published elsewhere in the meantime.

Please do not hesitate to contact us if you have any questions or would like to discuss these revisions further. We look forward to seeing the revised manuscript and thank you for the opportunity to review your work.

Best regards,

Olusegun Dada, PhD
Editorial Board Member
Communications Earth & Environment

EDITORIAL POLICIES AND FORMATTING

Editorial Policy: [Policy requirements](https://www.nature.com/documents/nr-editorial-policy-checklist.pdf) (Download the link to your computer as a PDF.)

- Behavioural and social science
- Ecological, evolutionary & environmental sciences
- Life sciences

<https://www.nature.com/documents/nr-reporting-summary.zip>

Furthermore, please align your manuscript with our format requirements, which are summarized on the following checklist: [Communications Earth & Environment formatting checklist](https://www.nature.com/documents/commsj-phys-style-formatting-checklist-article.pdf)

and also in our style and formatting guide [Communications Earth & Environment formatting guide](https://www.nature.com/documents/commsj-phys-style-formatting-guide-accept.pdf) .

***** DATA:** Communications Earth & Environment endorses the principles of the Enabling FAIR data project (<http://www.copdess.org/enabling-fair-data-project/>). We ask authors to make the data that support their conclusions available in permanent, publically accessible data repositories. (Please contact the editor if you are unable to make your data available).

All Communications Earth & Environment manuscripts must include a section titled "Data Availability" at the end of the Methods section or main text (if no Methods). More information on this policy, is available at <http://www.nature.com/authors/policies/data/data-availability-statements-data-citations.pdf>.

If a community resource is unavailable, data can be submitted to generalist repositories such as [figshare](https://figshare.com/) or [Dryad Digital Repository](http://datadryad.org/). Please provide a unique identifier for the data (for example a DOI or a permanent URL) in the data availability statement, if possible. If the repository does not provide identifiers, we encourage authors to supply the search terms that will return the data. For data that have been obtained from publically available sources, please provide a URL and the specific data product name in the data availability statement. Data with a DOI should be further cited in the methods reference section.

REVIEWER COMMENTS:

Reviewer #1 (Remarks to the Author):

Please add page and line numbers to any revised manuscript to help the reviewers out.

Throughout – consider using the term “tidal flat” versus “mudflat”. Tidal flat is more indicative of what the zone from MSL to MLLW really are outside of deltas. Many are sandy or even old coral flats.

P. 1. Point of clarification... Mangroves occur in tropical, sub-tropical, and warm temperate environments. Not exclusive to topics.

Introduction. Second to last paragraph. Very nice! It is very important to differentiate between small waves, large waves, and surge. All are different wave types hydrodynamically, as the authors clearly know. Most of what we see are normal wave models being applied to surge, and that is creating potentially disastrous expectations. True storm surge piles through many km's of mangroves without much effort until the land slope changes. Mangroves are great at wave suppression (sensu the old Wolanski and Furukawa work), but as the authors contend, maybe not as good with large waves and surge. So true.

Suggestion. Define up-front that storm surge (e.g., from cyclones) is not being modelled here. This statement is provided in the Discussion. I would also consider making this statement in some way in the Abstract. Could even delete reference to the number of data points in abstract. Not that important and articulated later.

P. 2. Data are plural, "...offshore wave data (from ERA-Interim) are considered..."

P.3. "...start of foreshore..." Add "of"

P. 3. Re: "For the mangrove greenbelt (part 3) an elevation (Vegz0) of 0 m +MSL and 0.5 m +MSL was assumed which is based on average distribution of mangroves with respect to inundation frequency". Very nice. As simple as this seems, past studies have not recognized that mangroves occur at MSL and above (mostly), which significantly affects results and interpretations. Modeling that way also takes away the tidal flats in the suppression scenarios, which are fully exposed for probably 25% of the time. My compliments for making the proper assumptions here. When I see a wave or surge suppression models and tidal flats are not included, it is probably time to stop reading the paper.

P. 4. Pay attention to citation contentions. Shifted here from Chicago to MLA.

Table 2. I would delete the last three rows. That information is well communicated in text and breaks up the matrix, which is very useful.

Equation 1. Is this the default equation used for SWAN, or is this derived by the investigators? I think the former. I may have missed the link but please make it clear. Readers will likely interrogate the formula more is self-derived versus well reviewed.

P. 6. Re: "Offshore the bottom friction was turned off to prevent unrealistic wave dissipation". It is unclear why this was done. Probably obvious, but perhaps explain why for those of us who do not use SWAN. That would exclude the locations of where coral reefs exist, right?

Results. Second line. Replace comma with period.

P. 7. First line starting with "Similarly,...". Suggest deleting the hyphens.

P. 7. Re: "Hence, the relative rate of energy dissipation decreases with the width of the forest." This sentence needs a slight tweak. The results can be interpreted both ways as written. Most readers will know what you mean but please make the message super clear.

Figure 4. Need to define "Bin" in the caption or delete the word in the figure. This is modeling jargon, I think.

Discussion, Second paragraph. Make it clear here again that this is for wave attenuation, not surge. I can see it now... Someone (a data miner v. reader) will say that modeling indicates that all we need is 100m to suppress storm surge. Unfortunately, mis-use of specific models has been an issue in the US and South America.

Discussion. Third paragraph. Suggesting that greenspace of 500m might work for surge. What is the source of these data? I don't buy it necessarily but could be convinced with proper modeling. I feel like the present modeling approach could work with a little more initialization. Remember though – surge waves come with a repetitive push of water, almost like a temporary seiche with the force constantly applied for 1-5 hours. It is not just normal wave propagation but a sustained build-up of water that is not allowed to drain because of the wind. I equate it more to a short-term river flood than a wave. Furthermore, what role does reduced atmospheric pressure play? When a cyclone hits, pressure has dropped significantly. Does this allow for greater water rise? This is not the case even for tsunamis. Can the authors work to address some of this as a way to caveat surge v. waves. At the very least, 500 m needs documenting, or needs additional modeling to offer.

Discussion. Second-to-last paragraph. Very nice! Also note that an important need is coupled modeling between coral reef and mangrove; perhaps before seawalls are suggested along tropical coastlines? I think that seawalls might be more justified in deltas where we are losing ground too rapidly to establish appropriate greenbelts. I used to disagree with that, but I am being increasingly convinced that in some areas, there is little option but hard structure (e.g., Louisiana).

Discussion. Last paragraph. A thought... Given that tidal flats are shown to play a significant role in attenuation, maybe say a bit about this result in summary here? Since mangroves mostly occur at MSL and higher, that means that at MLLW, the mudflat has a significant exposed area in front of the mangrove belt. But also, it is always there to baffle waves even at high tide. I would not under play the tidal flat results. That could be the biggest reason for citing your work.

Ref. 8. I know this might sound like self-interest, but instead of citing this non-peer-reviewed review, given that the entire review is based on two papers (modeling v. actual observations), it is better style to cite the original data. For this, that would be:

<https://doi.org/10.1672/07-232.1> and <https://doi.org/10.1016/j.ecss.2012.02.021>.

Furthermore, this paper should be incorporated into this manuscript. It adds relevance to your results.

<https://doi.org/10.1038/s41586-018-0805-8>

Consider adding the vastly different characteristics of mangrove forest development in future modeling scenarios. For example, at latitudinal extremes (e.g., coastal Louisiana; southern Australia, New Zealand). Figure 2, which I think is great, would be quite different for *Avicennia* advancing into marsh. It would be fun to see those wave attenuation scenarios. Would be of global relevance as greater tropicalization ensues.

/signed/

Ken Krauss, U.S. Geological Survey

Reviewer #2 (Remarks to the Author):

This manuscript quantifies the ability of mangrove forests to dissipate swell waves using an schematized numerical modelling approach. A large number of wave and forest conditions are simulated along 1D transects. Results are then compared to and set into context of mangrove forests widths on a global scale (including a useful meta-analysis of forest widths from around the globe).

Major comments:

In general the value of the work lies in the very large number of numerical simulations performed, rather than containing substantial conceptual advances. Thus, the authors could perhaps more carefully outline the novelty of the work, the links to existing theory, and state the critical advance.

The work would strongly benefit from a comparison of the results with the large body of existing literature in the area (either laboratory experiments or field observations), to help to set the work into context.

Other comments:

The title read somewhat strangely – why not simply use “Quantifying the width of ..” rather than quantifying confidence in. Also, at this point, the reader isn’t aware of what is meant by the effective width.

Abstract: Specify which data points? These are every point across multiple sites/transects? This seems an odd way to introduce the modelling as there’s no idea of resolution at this point.

Abstract: Independent of local conditions is vague –conditions of what?

Abstract: It would be good to put earlier that the work focusses on swell waves (rather than long or infragravity waves).

Introduction: A few examples of ‘They’ and ‘this’, ‘these’ – please add a noun to make it clear to what you are referring.

Introduction – refs 8, 9 15 – aren’t complete. Are these reports? Book chapters?

I suggest ‘was undertaken’ rather than ‘was done’

Methods – For surface area – do you mean frontal area (I think this is the more common term – see eg. Nepf 2012, ARFM)?

Methods – it would be good somewhere to explicitly note that the mangroves are generally emergent (except for the pioneers).

Methods – stating the number of points in total is odd – it would be better to frame the modelling in terms of the horizontal resolution along the transect.

Methods – can you justify why the use of drag formulations for willow trees are appropriate? Are there some statistics for tree size/characteristics than could be used to support this approach (especially given the majority of the reduction in wave height can be due to the mangrove roots rather than tree trunks for some species (see e.g. Maza et al. 2019, AWR))?

Methods – what is the Collins formulation for bottom friction (can this information be provided in supplementary material?)

Methods – Hs-Tm-01 – state what these parameters are.

Results – The lines/colors in the figures are about the worst possible combination for color-blind people. Can these be changed?

Results – figure 3 – given the main conclusion of the paper is to provide a width of forest required, it might make more sense to report these results in a dimensional sense in this context.

Results – figure 3 in caption state what the % is for on the figure (wave height reduction over foreshore).

Results – P8 – missing the word exceeds? (mangrove extent exceeds 100 meters in width).

Results – typo - 15.773 – should be 15,773.

Discussion – I think it would be more accurate to state that you have captured the range of wave attenuation, rather than the confidence, given these simulations were forced with different conditions.

Discussion - How do your dissipation results compare with previous field or laboratory measurements from previous literature? For example using dissipation length-scales from Henderson et al. 2017, CSR, or Maza et al 2022, Scientific Reports (who quantified reduction as a function of vegetation biomass).

Discussion

Below 500-> less than 500 m wide.

Communications Earth & Environment is committed to improving transparency in authorship. As part of our efforts in this direction, we are now requesting that all authors identified as 'corresponding author' create and link their Open Researcher and Contributor Identifier (ORCID) with their account on the Manuscript Tracking System prior to acceptance. ORCID helps the scientific community achieve unambiguous attribution of all scholarly contributions. You can create and link your ORCID from the home page of the Manuscript Tracking System by clicking on 'Modify my Springer Nature account' and following the instructions in the link below. Please also inform all co-authors that they can add their ORCIDs to their accounts and that they must do so prior to acceptance.

Version 1:

Decision Letter:

Dear Dr van Wesenbeeck,

Your manuscript titled "The importance of mangrove greenbelt width to reduce storm waves" has now been seen by our reviewers, whose comments appear below. In light of their advice we are delighted to say that we are happy, in principle, to publish a suitably revised version in Communications Earth & Environment.

We therefore invite you to revise your paper one last time to address the remaining concerns of our reviewers. At the same time we ask that you edit your manuscript to comply with our format requirements and to maximise the accessibility and therefore the impact of your work.

EDITORIAL REQUESTS:

*****Please take care to match our formatting and policy requirements. We will check revised manuscript and return manuscripts that do not comply. Such requests will lead to delays. *****

SUBMISSION INFORMATION:

OPEN ACCESS:

Communications Earth & Environment is a fully open access journal. Articles are made freely accessible on publication. For further information about article processing charges, open access funding, and advice and support from Nature Research, please visit <https://www.nature.com/commsenv/open-access>

Link Redacted

Best regards,

Alice Drinkwater, PhD
Associate Editor
Communications Earth & Environment

REVIEWERS' COMMENTS:

Reviewer #1 (Remarks to the Author):

I reviewed the revisions made to this manuscript, and I found the revision to be in excellent shape. I have two items to add for clarity,

(1) Re: "We also added something on planting mangroves on intertidal flats for restoration, which seems even more strange in light of the results of our paper." Please do not ever suggest planting mangroves on tidal flats seaward of existing mangroves. They do not belong there and will die.

(2) Re: DOI. You needed to take the last period off the DOI to get to the Zhang et al paper. Here it is,

<https://doi.org/10.1016/j.ecss.2012.02.021>

You are not missing much if you choose not to read it. Your modeling is better.

Nice job, and good luck with your continued research.

Ken Krauss

Reviewer #2 (Remarks to the Author):

The authors have done a good job of responding to my previous comments. (I will note that the line numbers in their response document in no way corresponded to the line numbers in the revised main manuscript document which was somewhat frustrating).

However, I noticed a few comments which were still missed:

L71 - I suggest changing 'was done' to 'was undertaken'

L221-225 - I still couldn't find a definition of Hs-Tm-01? Am I missing something? Hs is defined earlier as significant wave height, but I couldn't find Tm. Given Tz is peak period (i.e. different units), presumably the '-' isn't a minus sign?? These lines were quite confusing.

Reviewer #3 (Remarks to the Author):

The importance of mangrove greenbelt width to reduce storm waves:

The article employs a systematic approach, including data mining, selecting representative parameter combinations using the Maximum Dissimilarity Algorithm (MDA), extracting key mangrove tree characteristic values from extensive literature surveys, and conducting 216,000 model runs to comprehensively quantify the capacity of mangroves to reduce different types of waves. This systematic and comprehensive research method fully takes into account various biotic and abiotic factors in mangrove environments, as well as their interactions, thereby more accurately assessing the wave reduction effects of mangroves. It provides a scientific and rigorous research paradigm that can be referenced, helping to deeply understand the functioning mechanisms of complex ecosystems.

By executing a large number of model runs (216,000 times) and combining global data, the article conducts a detailed quantitative analysis of the wave reduction effects of mangrove belts of different widths, drawing statistically significant conclusions, such as mangrove belts wider than 500 meters can effectively dissipate more than 80% of the incoming wave energy. This research approach based on a large number of model runs and data analysis can effectively reduce conclusion biases caused by insufficient sample sizes or accidental factors, enhancing the reliability and universality of the research results. It provides more precise data support for coastal protection engineering design and mangrove conservation policy formulation, helping to achieve optimal resource allocation and protection effects.

In the model settings, a variety of practical factors are comprehensively considered, such as different types of mangroves (pioneer species, red mangroves, black mangroves), different tree densities (sparse, medium, dense), different foreshore slopes (1:500, 1:750, 1:1000), and different water level starting values (0.0 m + MSL, 0.5 m + MSL), making the model closer to real situations. This meticulous model setting reflects the researchers' profound understanding and respect for the complexity of mangrove ecosystems. By simulating the wave reduction process under different conditions, it can more accurately reflect the actual protective effects of mangroves under various natural and human disturbances, providing scientific basis for carrying out mangrove protection and restoration work in a location-specific manner, and helping to improve the targeting and effectiveness of protective measures.

The article clearly points out the relationship between mangrove belt width and wave reduction effects, especially emphasizing the significant advantages of mangrove belts wider than 500 meters in dissipating wave energy, as well as the uncertainty factors of wave reduction effects when the width is less than 500 meters. This clear conclusion provides important reference for the width design of mangrove belts in coastal protection engineering, helping to guide actual mangrove protection and restoration work, avoiding blindly pursuing too narrow or too wide mangrove belt widths, and achieving a balance between ecological protection and economic benefits. At the same time, it also provides scientific decision-making support for relevant decision-makers and managers, promoting the sustainable use of mangrove resources and the stable development of coastal areas.

The article analyzes the distribution of mangrove widths in different regions from a global perspective and puts forward targeted mangrove protection and restoration suggestions based on the research results, such as emphasizing the protection of larger areas of mangrove belts to avoid coastal instability, and considering the adoption of more scalable restoration methods, etc. This global perspective research and suggestions help enhance people's understanding of the important role of mangroves in global coastal protection, promote exchanges and cooperation among countries, and jointly address coastal protection challenges. At the same time, it also provides macro-level guiding ideas for global mangrove protection and restoration work, promoting the comprehensive improvement of mangrove ecosystem service functions and enhancing the ecological security and sustainable development capabilities of coastal areas.

From the discussion section, here are 3 suggestions:

1. Expand the scope of research to include more extreme marine conditions

Suggestion: Future research should incorporate storm surges, infragravity waves, and typhoon conditions into the study of mangroves' protective effects. These extreme marine conditions pose significant threats to coastal areas, and mangroves may have different levels of effectiveness in mitigating their impacts compared to normal wave conditions. By conducting simulations and field observations under these extreme scenarios, a more comprehensive understanding of mangroves' overall coastal protection capabilities can be achieved. This will provide more robust scientific evidence for designing coastal protection strategies that fully harness the potential of mangroves in various situations.

2. Develop mangrove-specific KC-CD relationships for extreme hydrodynamic conditions

Suggestion: Efforts should be made to establish KC-CD (Keulegan-Carpenter number - drag coefficient) relationships specifically for mangroves under extreme hydrodynamic conditions. Currently, the study uses the KC-CD relationship derived for willows as a substitute, which may introduce errors due to differences in morphology and structure between willows and mangroves. Conducting more field experiments and laboratory simulations to collect data on mangroves' resistance characteristics under high water levels, large waves, and other extreme conditions will enable the development of accurate KC-CD relationships for mangroves. This will enhance the precision and applicability of numerical models used to simulate wave attenuation by mangroves, leading to more reliable assessments of their coastal protection functions.

3. Optimize the structure and combine protection measures for narrow mangrove belts

Suggestion: For mangrove belts narrower than 500 meters, research should focus on optimizing their structure to improve wave attenuation capacity. This could involve increasing tree density, improving tree species combinations, and enhancing the overall health and resilience of the mangrove ecosystem within these narrow belts. Additionally, exploring hybrid

protection solutions that integrate narrow mangrove belts with artificial coastal defense structures, such as levees or seawalls, can be a practical approach. These hybrid solutions can leverage the advantages of both natural vegetation and engineered structures to provide more effective and reliable coastal protection, especially in areas where wider mangrove belts are not feasible due to coastal topography or human activities.

Reply to reviewers

REVIEWER COMMENTS:

The authors would like to thank all three reviewers for their valuable comments and insights. Their constructive tone of voice was really appreciated, and the modifications made the paper hopefully clearer and more relevant.

Reviewer #1 (Remarks to the Author):

Please add page and line numbers to any revised manuscript to help the reviewers out.

Apologies for this. Must have been very inconvenient. Line and page numbers are added now.

Throughout – consider using the term “tidal flat” versus “mudflat”. Tidal flat is more indicative of what the zone from MSL to MLLW really are outside of deltas. Many are sandy or even old coral flats.

Good remark. I do like that better. Changed throughout the paper.

P. 1. Point of clarification... Mangroves occur in tropical, sub-tropical, and warm temperate environments. Not exclusive to topics.

Added subtropical in line 31-32

Introduction. Second to last paragraph. Very nice! It is very important to differentiate between small waves, large waves, and surge. All are different wave types hydrodynamically, as the authors clearly know. Most of what we see are normal wave models being applied to surge, and that is creating potentially disastrous expectations. True storm surge piles through many km's of mangroves without much effort until the land slope changes. Mangroves are great at wave suppression (sensu the old Wolanski and Furukawa work), but as the authors contend, maybe not as good with large waves and surge. So true.

Thank you for sharing this point of view. We are currently working on improving prediction of mangrove influence on surges through global 2D modelling as we agree that current work may overestimate the capacities of mangroves to reduce surges.

Suggestion. Define up-front that storm surge (e.g., from cyclones) is not being modelled here. This statement is provided in the Discussion. I would also consider making this statement in some way in the Abstract. Could even delete reference to the number of data points in abstract. Not that important and articulated later.

Clear point. We deleted the number of data points and added “for wind and swell waves” in sentence 19 of the abstract

P. 2. Data are plural, "...offshore wave data (from ERA-Interim) are considered..."
Line 46, changed 'is' to 'are'

P.3. "...start of foreshore..." Add "of"

Line 6 added 'the', line 8 added 'of the'

P. 3. Re: "For the mangrove greenbelt (part 3) an elevation (Vegz0) of 0 m +MSL and 0.5 m +MSL was assumed which is based on average distribution of mangroves with respect to inundation frequency". Very nice. As simple as this seems, past studies have not recognized that mangroves occur at MSL and above (mostly), which significantly affects results and interpretations. Modeling that way also takes away the tidal flats in the suppression scenarios, which are fully exposed for probably 25% of the time. My compliments for making the proper assumptions here. When I see a wave or surge suppression models and tidal flats are not included, it is probably time to stop reading the paper.

Thank you for the positive feedback. Your remarks are supported by the results of our modelling exercise illustrating the importance of the tidal flats in damping the highest waves before they enter the forest.

P. 4. Pay attention to citation contentions. Shifted here from Chicago to MLA.

Noted and adapted. See line 14.

Table 2. I would delete the last three rows. That information is well communicated in text and breaks up the matrix, which is very useful.

If it is clear from the text than that is perfect. Deleted.

Equation 1. Is this the default equation used for SWAN, or is this derived by the investigators? I think the former. I may have missed the link but please make it clear. Readers will likely interrogate the formula more is self-derived versus well reviewed.
Agree, changed text to clarify and included the reference to the Mendez and Losada paper from 2004. Page 6, line 8-11.

P. 6. Re: "Offshore the bottom friction was turned off to prevent unrealistic wave dissipation". It is unclear why this was done. Probably obvious, but perhaps explain why for those of us who do not use SWAN. That would exclude the locations of where coral reefs exist, right?

We added some text to clarify "Offshore the bottom friction was turned off to prevent unrealistic wave dissipation before the wave entered the synthetic tidal flat profile."

Coral reefs are not specifically included in this exercise unless they are recorded in the global data bottom profile. Then their initial depth is taken to start the tidal flat profile.

Results. Second line. Replace comma with period.

Done

P. 7. First line starting with “Similarly,...”. Suggest deleting the hyphens.

Done

P. 7. Re: “Hence, the relative rate of energy dissipation decreases with the width of the forest.” This sentence needs a slight tweak. The results can be interpreted both ways as written. Most readers will know what you mean but please make the message super clear.

Changed in: “Hence, the relative amount of energy reduction per meter gets lower once the wave already travelled through the forest for a considerable distance and is already partly dampened. “

Figure 4. Need to define “Bin” in the caption or delete the word in the figure. This is modeling jargon, I think.

The figure is adapted

Discussion, Second paragraph. Make it clear here again that this is for wave attenuation, not surge. I can see it now... Someone (a data miner v. reader) will say that modeling indicates that all we need is 100m to suppress storm surge. Unfortunately, mis-use of specific models has been an issue in the US and South America.

Line 27, page 9, included (wind and swell waves). Note that whole paragraph on mangroves versus surges and storm waves follows below.

Discussion. Third paragraph. Suggesting that greenspace of 500m might work for surge. What is the source of these data? I don't buy it necessarily but could be convinced with proper modeling.

Seems there is a slight misunderstanding here in our text. I modified line 35-38 of page 9 to:

“Nevertheless, wider green belts in combination with gradually sloping tidal flats are more effective with more confidence to dampen waves. However, in many places these cannot realistically be conserved or restored due to limits in coastal topography on the front and encroachment of anthropogenic activities in the back. “

I feel like the present modeling approach could work with a little more initialization. Remember though – surge waves come with a repetitive push of water, almost like a temporary seiche with the force constantly applied for 1-5 hours. It is not just normal wave propagation but a sustained build-up of water that is not allowed to drain because of the wind. I equate it more to a short-term river flood than a wave. Furthermore, what role does reduced atmospheric pressure play? When a cyclone hits, pressure has dropped significantly. Does this allow for greater water rise? This is not the case even for tsunamis. Can the authors work to address some of this as a way to caveat surge v. waves. At the very least, 500 m needs documenting, or needs additional modeling to offer.

Thank you for these thoughts. These questions were also some of our follow-up questions

from this paper. With surges until now only a 1D approach has been used in a global context (see Menendez et al.). However, the way that a surge spreads is very much dependent on 2D geomorphology and channelization of mangrove forests (See Pelckmans et al.). Therefore, we already executed a first 2D global modelling exercise using a new delta DM topography. We expect to submit this work beginning of next year.

Discussion. Second-to-last paragraph. Very nice! Also note that an important need is coupled modeling between coral reef and mangrove; perhaps before seawalls are suggested along tropical coastlines? I think that seawalls might be more justified in deltas where we are losing ground too rapidly to establish appropriate greenbelts. I used to disagree with that, but I am being increasingly convinced that in some areas, there is little option but hard structure (e.g., Louisiana).

I would like to continue this discussion at some point. I think seawalls can be justified in areas with high population pressure. However, sometimes we need some erosion or managed retreat to make space for mangroves and marshes again, especially with rising sea levels. I think and hope we will put more effort in learning about the combination of green and grey approaches in the future.

Discussion. Last paragraph. A thought... Given that tidal flats are shown to play a significant role in attenuation, maybe say a bit about this result in summary here? Since mangroves mostly occur at MSL and higher, that means that at MLLW, the mudflat has a significant exposed area in front of the mangrove belt. But also, it is always there to baffle waves even at high tide. I would not under play the tidal flat results. That could be the biggest reason for citing your work.

Good point. We also added something on planting mangroves on intertidal flats for restoration, which seems even more strange in light of the results of our paper.

Ref. 8. I know this might sound like self-interest, but instead of citing this non-peer-reviewed review, given that the entire review is based on two papers (modeling v. actual observations), it is better style to cite the original data. For this, that would be: <https://doi.org/10.1672/07-232.1> and <https://doi.org/10.1016/j.ecss.2012.02.021>.

Thank you, we validated our surge model for the next paper with your data. We included it here. The last DOI is not working, so we are not sure what you are referring to.

Furthermore, this paper should be incorporated into this manuscript. It adds relevance to your results. <https://doi.org/10.1038/s41586-018-0805-8>

True. Belongs here indeed. Added on page 9 line 13 and line 25.

Consider adding the vastly different characteristics of mangrove forest development in future modeling scenarios. For example, at latitudinal extremes (e.g., coastal Louisiana; southern Australia, New Zealand). Figure 2, which I think is great, would be quite different for Avicennia advancing into marsh. It would be fun to see those wave attenuation scenarios. Would be of global relevance as greater tropicalization ensues. Good suggestion for future work. Maybe interesting for stunted mangroves in Louisiana functioning. Hijuelos et al. in PLOS. <https://doi.org/10.1371/journal.pone.0216695> We are continuing to work on phenological plasticity of mangroves globally.

/signed/ Ken Krauss, U.S. Geological Survey

Thank you for your thoughts and constructive comments. Best regards, Bregje van Wesenbeeck

Reviewer #2 (Remarks to the Author):

This manuscript quantifies the ability of mangrove forests to dissipate swell waves using a schematized numerical modelling approach. A large number of wave and forest conditions are simulated along 1D transects. Results are then compared to and set into context of mangrove forests widths on a global scale (including a useful meta-analysis of forest widths from around the globe).

Major comments:

In general the value of the work lies in the very large number of numerical simulations performed, rather than containing substantial conceptual advances. Thus, the authors could perhaps more carefully outline the novelty of the work, the links to existing theory, and state the critical advance.

We agree that this paper conceptually does not offer anything specifically new. However, by making use of a combination of methods and techniques we were able to generate a unique modelling study at scales that are unrepresented. The vast amount of data that we are able to generate, which is largely taken from global data sources and not synthetically generated, sheds new light on mangrove widths and functioning in relation to current literature and policy conceptions. Furthermore, it provides a sense of uncertainty levels in modelling studies depending on mangrove width. Something that was not specifically highlighted before but has implications for how mangroves and their capacity to dampen waves can be relied upon. To be specific, caution should be taken with small mangrove belts as their capacity to reduce waves is dependent on a lot of local factors. To reflect this better in the paper and to link it better to current work we have considerably modified line 10-15 on page 2. We hope this described better what we add to the vast amount of literature that is already present on this topic.

The work would strongly benefit from a comparison of the results with the large body of existing literature in the area (either laboratory experiments or field observations), to help to set the work into context.

Agreed, we again surveyed the large body of work in this field. We cited reviews that offer an excellent overview of most of the work. First, the early work of Hashim et al. 2013, providing an overview of greenbelt policies and laboratory and field work on mangroves and wave attenuation. Second the paper of van Hespén et al. 2021.

We added several references to better embed our model observations in the literature, for example page 8, line 4, referring to Dalrymple 1984, Koch et al 2009, Bao 2011 and Lee et al 2021.

To relate to other global studies we already included the following references: Temmerman et al. 2023, Menendez et al. 2020, van Zelst et al. 2021, Tiggeloven et al. 2022.

For a more comprehensive overview of vegetation modeling we added

Other comments:

The title read somewhat strangely – why not simply use “Quantifying the width of ..” rather than quantifying confidence in. Also, at this point, the reader isn’t aware of what is meant by the effective width.

Good comment. We revised the title to: The importance of mangrove greenbelt width of mangrove green belts to reduce storm waves.

Abstract: Specify which data points? These are every point across multiple sites/transects? This seems an odd way to introduce the modelling as there’s no idea of resolution at this point.

Agreed, we deleted the number of data points from the abstract in line 18.

Abstract: Independent of local conditions is vague –conditions of what?

Line 25, changed into “independent of wave and forest characteristics”

Abstract: It would be good to put earlier that the work focusses on swell waves (rather than long or infragravity waves).

Agreed, added in line 19 and in line 26.

Introduction: A few examples of ‘They’ and ‘this’, ‘these’ – please add a noun to make it clear to what you are referring.

Agreed, modified line 39-40, 47 by changing wording and deleting ‘they’.

Introduction – refs 8, 9 15 – aren’t complete. Are these reports? Book chapters?

Those references are review reports. However, we changed these to peer-reviewed literature that is more recent by adding in the two reviews of Hashim et al. 2013 and van Hespén et al. 2021.

I suggest ‘was undertaken’ rather than ‘was done’

Changed sentence 10-11 of page 2.

Methods – For surface area – do you mean frontal area (I think this is the more common term – see eg. Nepf 2012, ARFM)?

Agreed, we changed the term in sentence 12 and 16 on page 4

Methods – it would be good somewhere to explicitly note that the mangroves are generally emergent (except for the pioneers).

Page 5, sentence 1, added ‘relative to water level’. And added text in line 1 and 2.

Methods – stating the number of points in total is odd – it would be better to frame the modelling in terms of the horizontal resolution along the transect.

Number of output points are deleted from the table but kept in the text s those are the data points that are used to compose figure 4. Resolution is added in line 26 and 27 of page 5

Methods – can you justify why the use of drag formulations for willow trees are appropriate? Are there some statistics for tree size/characteristics than could be used to support this approach (especially given the majority of the reduction in wave height can be due to the mangrove roots rather than tree trunks for some species (see e.g. Maza et al. 2019, AWR))?

Agree, this is a difficult issue and there is limited information to base Cd values on. None of the options is perfect. Based on available data and information we can either include Cd values that have been derived for mangroves under benign conditions from the field or we use drag coefficients from scales flume studies with plastic or we include drag coefficients that were derived for more extreme conditions for willows. As most vegetation in models is scaled with artificial structures, we thought willows to be a better proxy for a mangrove than steel wires or plastic strips. In addition, drag coefficients that are based on benign conditions may largely overestimate effects of trees under extreme conditions (see papers by van Wesenbeeck et al. 2022 and Kalloe et al, 2024). Hence, we here choose for conservative Cd estimates to not overestimate mangrove effects. In addition, we applied different Cds for branches, trunks and roots. We agree that influence of roots will be underestimated in willow trees. Hence, again, here we choose for a conservative measure. However, overall other studies do not include canopies, and the willow study does.

Methods – what is the Collins formulation for bottom friction (can this information be provided in supplementary material?)

In SWAN there are different methods to include bottom friction. For our purpose the Collins bottom friction coefficient can be adapted for the mudflat and within the forest to represent litter and increased roughness. We have added a reference to the paper of Collins. Line 32.

Methods – Hs-Tm-01 – state what these parameters are.

Changed, line 13+14

Results – The lines/colors in the figures are about the worst possible combination for color-blind people. Can these be changed?

Colors of all figures are changed taking into account various types of color blindness.

Results – figure 3 – given the main conclusion of the paper is to provide a width of forest required, it might make more sense to report these results in a dimensional sense in this context.

The aim of this figure is to illustrate the importance of the tidal flat in damping waves before they enter the forest.

Results – figure 3 in caption state what the % is for on the figure (wave height reduction over foreshore).

We adapted the figure and the caption.

Results – P8 – missing the word exceeds? (mangrove extent exceeds 100 meters in width).

Page 8, line 21-20 adapted.

Results – typo - 15.773 – should be 15,773.

Changed

Discussion – I think it would be more accurate to state that you have captured the range of wave attenuation, rather than the confidence, given these simulations were forced with different conditions.

Page 9, line 8 and 9. True, not exactly confidence, but replacing with range leads to a sentence that is not correct. Now changed in uncertainty.

Discussion - How do your dissipation results compare with previous field or laboratory measurements from previous literature? For example using dissipation length-scales from Henderson et al. 2017, CSR, or Maza et al 2022, Scientific Reports (who quantified reduction as a function of vegetation biomass).

We explored a large set of conditions and possible parameter values based on previous literature and on global data. Hence, most studies will fit within the band width of our results. For the two specific studies mentioned above this is also the case. However, please note that both studies above were executed with low water levels in very dense canopies, resulting in Cd values that are twice as high as Cds reported for waves with higher water levels and higher waves (Cd = 2 for Henderson). Hence. We choose our values more conservative by using Cd from willows that were obtained under more extreme conditions (Cd < 1). For Henderson 100 meter into the forest was enough to attenuate waves largely. This is illustrated in our figure 4. In certain cases this attenuation is possible. However for a 100 meter wide forest once waves are higher and densities are lower, and hence, Cd coefficients are lower, attenuation can as well be only 10 % in 100 metres. Illustrating the message of our paper, that for small forest width attenuation completely depends on local conditions and cannot be extrapolated to other situations. The Maza study was executed on salt marsh vegetation, which is inherently different compared to woody mangrove vegetation.

Discussion

Below 500-> less than 500 m wide.

Changed, page 9, line 31

Reviewer 3

This is an interesting study, but I have some questions that I hope will help with the manuscript.

Model Calibration and Validation Details: The paper mentions the use of data to calibrate the wave dissipation model, but does not provide detailed information on the source of these experimental data, the experimental design, and the specific steps and results of the calibration process. Could you please provide more detailed information regarding the model calibration and validation? This includes, but is not limited to, a description of the experimental dataset, the specific methods used during calibration, and a comparison of the model's performance before and after calibration. This information is crucial for assessing the accuracy and reliability of the model.

The original wave – vegetation models are based on schematizations by Mendez and Losada (2004), the layer schematization which is implemented and tested by Suzuki et al. This was better explained in the text on page 6, line 8 till 11.

Sufficiency of Simulation Under Extreme Conditions: The paper discusses water levels and wave conditions for different return periods, but the simulation of extreme climatic events (such as tropical storms and typhoons) seems limited. Have you considered the impact of these extreme events on the wave attenuation capability of mangroves? If so, please elaborate on the methods used to simulate these extreme conditions and the results obtained. If not, please discuss why these extreme scenarios were not included and explain the potential implications this may have on the study's conclusions.

Extreme conditions are in principle included as the offshore data is a re-analysis of Era Interim data with tropical storm included. However, due to relatively small spatial scales that typhoons exhibit effects and the coarse resolution of global data and modelling effects of storms may be slightly underestimated in the data. We have added brief text for clarification in line 25 on page 3.

Potential Impact of Hybrid Defense Systems: The paper primarily focuses on the wave attenuation effect of mangroves alone, but in practical applications, mangroves are often used in conjunction with other coastal defense structures (such as seawalls and breakwaters). Have you assessed the potential impact of such hybrid defense systems on wave attenuation? If not, it is recommended that you explore the synergistic effects of these hybrid systems and analyze their potential contributions to enhancing coastal protection capabilities. This not only enhances the practicality of the paper but also helps to provide a more comprehensive assessment of the role of mangroves in coastal defense.

Good suggestion, we already performed and published an analysis of potential of hybrid systems in the paper by van Zelst et al. 2021 that is referenced in line 33 on page 1. Hence, we thought this is out of scope for the current paper.

REVIEWERS' COMMENTS:

Reviewer #1 (Remarks to the Author):

I reviewed the revisions made to this manuscript, and I found the revision to be in excellent shape. I have two items to add for clarity,

(1) Re: "We also added something on planting mangroves on intertidal flats for restoration, which seems even more strange in light of the results of our paper." Please do not ever suggest planting mangroves on tidal flats seaward of existing mangroves. They do not belong there and will die.

Reply: We are very aware of that and that is by no means suggested. We modified text at the end of the discussion to be clear on this. P7 line 15- 22

(2) Re: DOI. You needed to take the last period off the DOI to get to the Zhang et al paper. Here it is,

<https://doi.org/10.1016/j.ecss.2012.02.021>

You are not missing much if you choose not to read it. Your modeling is better.

Reply: We are aware of this paper, but as it deals with surges we did not include it here. We will in our future work.

Nice job, and good luck with your continued research.

Ken Krauss

Reviewer #2 (Remarks to the Author):

The authors have done a good job of responding to my previous comments. (I will note that the line numbers in their response document in no way corresponded to the line numbers in the revised main manuscript document which was somewhat frustrating).

Reply: I apologize. It seems page and line numbers changed or moved down. We appreciate your rigorous reviewing. Thank you.

However, I noticed a few comments which were still missed:
L71 - I suggest changing 'was done' to 'was undertaken'

Reply: this is now on Page 2, line 20, Done

L221-225 - I still couldn't find a definition of Hs-Tm-01? Am I missing something? Hs is defined earlier as significant wave height, but I couldn't find Tm. Given Tz is peak period (i.e. different units), presumably the '-' isn't a minus sign?? These lines were quite confusing.

Reply: Indeed not a minus sign. I removed the sign and added first the description Hs is average Wave height Tm01 is the mean average wave period. See Line 14-15 page 5

Reviewer #3 (Remarks to the Author):

The importance of mangrove greenbelt width to reduce storm waves:

The article employs a systematic approach, including data mining, selecting representative parameter combinations using the Maximum Dissimilarity Algorithm (MDA), extracting key mangrove tree characteristic values from extensive literature surveys, and conducting 216,000 model runs to comprehensively quantify the capacity of mangroves to reduce different types of waves. This systematic and comprehensive research method fully takes into account various biotic and abiotic factors in mangrove environments, as well as their interactions, thereby more accurately assessing the wave reduction effects of mangroves. It provides a scientific and rigorous research paradigm that can be referenced, helping to deeply understand the functioning mechanisms of complex ecosystems.

By executing a large number of model runs (216,000 times) and combining global data, the article conducts a detailed quantitative analysis of the wave reduction effects of mangrove belts of different widths, drawing statistically significant conclusions, such as mangrove belts wider than 500 meters can effectively dissipate more than 80% of the incoming wave energy. This research approach based on a large number of model runs and data analysis can effectively reduce conclusion biases caused by insufficient sample sizes or accidental factors, enhancing the reliability and universality of the research results. It provides more precise data support for coastal protection engineering design and mangrove conservation policy formulation, helping to achieve optimal resource allocation and protection effects.

In the model settings, a variety of practical factors are comprehensively considered, such as different types of mangroves (pioneer species, red mangroves, black mangroves), different tree densities (sparse, medium, dense), different foreshore slopes (1:500, 1:750, 1:1000), and different water level starting values (0.0 m + MSL, 0.5 m + MSL), making the model closer to real situations. This meticulous model setting reflects the researchers' profound understanding and respect for the

complexity of mangrove ecosystems. By simulating the wave reduction process under different conditions, it can more accurately reflect the actual protective effects of mangroves under various natural and human disturbances, providing scientific basis for carrying out mangrove protection and restoration work in a location-specific manner, and helping to improve the targeting and effectiveness of protective measures.

The article clearly points out the relationship between mangrove belt width and wave reduction effects, especially emphasizing the significant advantages of mangrove belts wider than 500 meters in dissipating wave energy, as well as the uncertainty factors of wave reduction effects when the width is less than 500 meters. This clear conclusion provides important reference for the width design of mangrove belts in coastal protection engineering, helping to guide actual mangrove protection and restoration work, avoiding blindly pursuing too narrow or too wide mangrove belt widths, and achieving a balance between ecological protection and economic benefits. At the same time, it also provides scientific decision-making support for relevant decision-makers and managers, promoting the sustainable use of mangrove resources and the stable development of coastal areas.

The article analyzes the distribution of mangrove widths in different regions from a global perspective and puts forward targeted mangrove protection and restoration suggestions based on the research results, such as emphasizing the protection of larger areas of mangrove belts to avoid coastal instability, and considering the adoption of more scalable restoration methods, etc. This global perspective research and suggestions help enhance people's understanding of the important role of mangroves in global coastal protection, promote exchanges and cooperation among countries, and jointly address coastal protection challenges. At the same time, it also provides macro-level guiding ideas for global mangrove protection and restoration work, promoting the comprehensive improvement of mangrove ecosystem service functions and enhancing the ecological security and sustainable development capabilities of coastal areas.

From the discussion section, here are 3 suggestions:

1. Expand the scope of research to include more extreme marine conditions
Suggestion: Future research should incorporate storm surges, infragravity waves, and typhoon conditions into the study of mangroves' protective effects. These extreme marine conditions pose significant threats to coastal areas, and mangroves may have different levels of effectiveness in mitigating their impacts compared to normal wave conditions. By conducting simulations and field observations under these extreme scenarios, a more comprehensive understanding of mangroves' overall coastal protection capabilities can be achieved. This will provide more robust scientific evidence for designing coastal protection strategies that fully harness the potential of mangroves in various situations.

2. Develop mangrove-specific KC-CD relationships for extreme hydrodynamic conditions

Suggestion: Efforts should be made to establish KC-CD (Keulegan-Carpenter number - drag coefficient) relationships specifically for mangroves under extreme hydrodynamic conditions. Currently, the study uses the KC-CD relationship derived for willows as a substitute, which may introduce errors due to differences in morphology and structure between willows and mangroves. Conducting more field experiments and laboratory simulations to collect data on mangroves' resistance characteristics under high water levels, large waves, and other extreme conditions will enable the development of accurate KC-CD relationships for mangroves. This will enhance the precision and applicability of numerical models used to simulate wave attenuation by mangroves, leading to more reliable assessments of their coastal protection functions.

3. Optimize the structure and combine protection measures for narrow mangrove belts

Suggestion: For mangrove belts narrower than 500 meters, research should focus on optimizing their structure to improve wave attenuation capacity. This could involve increasing tree density, improving tree species combinations, and enhancing the overall health and resilience of the mangrove ecosystem within these narrow belts. Additionally, exploring hybrid protection solutions that integrate narrow mangrove belts with artificial coastal defense structures, such as levees or seawalls, can be a practical approach. These hybrid solutions can leverage the advantages of both natural vegetation and engineered structures to provide more effective and reliable coastal protection, especially in areas where wider mangrove belts are not feasible due to coastal topography or human activities.

Reply: Thank you for time and for your elaborate review. We are happy to hear you find our paper of importance and that the main messages come across well. We appreciate the valuable suggestions for future research. Suggestion 1 is briefly included in page 7 line 1-5. Suggestion 3 is briefly captured in line 5-12 on page 7. For the second suggestion, we agree, but we did not include this in the current paper. However, we already performed large wave basin test with mangroves in May 2024 that may allow us to derive the relationships you suggest. We will take these suggestions with us for future research.

This is an interesting study, but I have some questions that I hope will help with the manuscript.

Model Calibration and Validation Details: The paper mentions the use of data to calibrate the wave dissipation model, but does not provide detailed information on the source of these experimental data, the experimental design, and the specific steps and results of the calibration process. Could you please provide more detailed information regarding the model calibration and validation? This includes, but is not limited to, a description of the experimental dataset, the specific methods used during calibration, and a comparison of the model's performance before and after calibration. This information is crucial for assessing the accuracy and reliability of the model.

Sufficiency of Simulation Under Extreme Conditions: The paper discusses water levels and wave conditions for different return periods, but the simulation of extreme climatic events (such as tropical storms and typhoons) seems limited. Have you considered the impact of these extreme events on the wave attenuation capability of mangroves? If so, please elaborate on the methods used to simulate these extreme conditions and the results obtained. If not, please discuss why these extreme scenarios were not included and explain the potential implications this may have on the study's conclusions.

Potential Impact of Hybrid Defense Systems: The paper primarily focuses on the wave attenuation effect of mangroves alone, but in practical applications, mangroves are often used in conjunction with other coastal defense structures (such as seawalls and breakwaters). Have you assessed the potential impact of such hybrid defense systems on wave attenuation? If not, it is recommended that you explore the synergistic effects of these hybrid systems and analyze their potential contributions to enhancing coastal protection capabilities. This not only enhances the practicality of the paper but also helps to provide a more comprehensive assessment of the role of mangroves in coastal defense.